# The Impact of Hoffmann Reflex on Standing Postural Control Complexity in the Elderly with Impaired Plantar Sensation

**DOI:** 10.3390/e25010064

**Published:** 2022-12-29

**Authors:** Mengzi Sun, Fangtong Zhang, Kelsey Lewis, Qipeng Song, Li Li

**Affiliations:** 1School of Sports Science and Physical Education, Nanjing Normal University, Nanjing 210023, China; 2Department of Health Sciences and Kinesiology, Georgia Southern University, Statesboro, GA 30458, USA; 3Biomechanics Laboratory, Beijing Sport University, Beijing 100084, China; 4Biomechanics Laboratory, Shandong Sport University, Jinan 276826, China

**Keywords:** neuropathy, H-reflex, postural control, balance, complexity

## Abstract

In people with peripheral neuropathy (PN), impaired plantar sensation can cause adaptive changes in the central nervous system (CNS), resulting in changes in the standing postural control, which is reflected in the variability of standing output signals. Standard deviation (SD) and entropy are reliable indicators of system variability, especially since entropy is highly sensitive to diseased populations. The relation between SD and entropy, CNS and center of pressure (COP) variability is unclear for people with severe PN. The purpose of this study was to explore the adaptability of the CNS to the severe of PN and its effect on the degree and complexity of COP variability. Here, people with PN were divided into less affected (LA) and more affected (MA) groups based on plantar pressure sensitivity. We studied Hoffmann reflex (H-reflex) and standing balance performance with the control group (n = 8), LA group (n = 10), and MA group (n = 9), recording a 30 s COP time series (30,000 samples) of double-leg standing with eyes open. We observed that the more affected group had less COP complexity than people without PN. There is a significant negative correlation between the SD and sample entropy in people without PN, less affected and more affected. The COP complexity in people without PN was inversely correlated with H-reflex. We concluded that: (1) The complexity of COP variability in patients with severe plantar sensory impairment is changed, which will not affect the degree of COP variability; (2) The independence of the COP entropy in the AP and ML directions decreased, and the interdependence increased in people with PN; (3) Although the CNS of people with PN has a greater contribution to standing balance, its modulation of standing postural control is decreased.

## 1. Introduction

Fall risks are one of the main factors that threaten the health and life of older adults [1]. Decreased postural control in people with peripheral neuropathy (PN) is a major cause of falls [2,3]. PN is a common degenerative disease caused by peripheral nerve damage in older adults [4]. The development of PN disease is characterized by a gradual distal to proximal development [5]. The main symptoms of PN are plantar paresthesia and slowed nerve conduction, which stems from damage to the lower extremities’ sensory afferent pathways [6]. Impaired or lost plantar sensation is one of the main factors of reduced postural control in people with PN [7].

Postural control mechanism changes among people with PN with the progression of the disease. A previous study showed that type I and II afferents are essential for balance- and gait-related postural control. The smaller type II (tactile feedback) is particularly critical for maintaining balance [8]. People with PN rely more on type I afferents to compensate for their decreased tactile sensitivity [8,9,10]. The severity of people with PN have different degrees of plantar sensory impairment, the contribution of tactile feedback in maintaining balance is different, and the compensatory effects of type I afferents also vary.

The modulation of the central nervous system (CNS) changes with severity in people with PN. Maintaining balance relies mainly on the plantar sensory (type II afferent pathway) among people without PN. However, people with PN have impaired type II afferent pathway, so they depend more on the type I afferent pathway. Hoffmann-reflex (H-reflex) can be used to examine the function of Ia afferent pathway and the effects of CNS modulation. Our previous study observed that the severity of PN affected posture modulations [10]. The H-reflex decreased when tested in prone compared to standing posture in healthy people, but this posture-related change diminished in people with severe PN. These results indicated that type I afferent fiber reflex loops (H-reflex) exert greater control over standing posture in people with severely impaired plantar sensitivity to compensate for plantar numbness.

Postural control is determined by the combined contribution of sensory, central, and motor pathways [11]. So, sensory impairment and CNS modulation changes can affect postural control in people with severe PN. Meyer and coworkers [12] reduced plantar sensory input through anesthesia. They reported that the center of pressure (COP) velocity was significantly increased after plantar anesthesia but had no effect on the range of movement of the COP. However, Hong and coworkers observed a different result [13]. They further studied that ice-induced plantar desensitization only affected the degree of COP variability and did not change the complexity of the time series. These studies indicated that traditional COP indicators, such as velocity, may not be related to the complexity of the COP [14]. Complexity in this project is defined as the complexity of physiological systems, using entropy to assess the regularity of the point-to-point fluctuations in the AP and ML of COP time series [15]. Traditional COP indicators have limitations, and the random fluctuations of COP at the spatial scale need to be reflected by the variability of COP. The COP variability can be used to evaluate postural control in people with PN. Metrics to quantify COP variability are often expressed in terms of the degree of variation (standard deviation) and complexity (entropy) [16]. Standard deviation is a typical linear calculation used to describe dynamic variability [17]. Entropy has been used to evaluate the complexity of dynamical systems [18,19]. There is a correlation between standard deviation and entropy because both are used to evaluate variability. Manor et al. investigated that 6, 12, 18, and 24 weeks of Tai Chi training increased the complexity of COP and was associated with improved plantar sensitivity [11]. These results suggest that greater COP complexity indicates enhanced posture control in people with PN. From these reports, we speculate that as the PN progresses, the postural control of people with PN decreases, manifested as a decrease in COP complexity.

As the severity of PN increases, the degree of plantar sensory impairment increases, resulting in changes in the CNS modulation of type I and type II afferent pathways, affecting postural control. However, the impact of changes in CNS modulation caused by the severity of PN disease on postural control is currently unknown. Exploring the effect of CNS modulation changes caused by the severity of PN on postural control helps understand the modulation mechanism of postural control.

We aimed to explore the CNS adaptability to the severity of PN and its effect on the degree and complexity of COP variability. The research hypotheses were: (1) With the increase in PN severity, the degree and the complexity of COP variability decreased, and the two are highly correlated; (2) The degree and complexity of COP variability in healthy people are negatively correlated with H-reflex. In contrast, the degree and the complexity of COP variability are positively correlated with H-reflex in people with PN.

## 2. Materials and Methods


**Experiment procedure**


Twenty-seven participants over 65 years old were included in this study, including people with intact and deficit plantar tactile sensitivity. For safety reasons, people with the following conditions were excluded from this study: (1) heart condition; (2) high blood pressure; (3) spinal cord disease; (4) losing balance because of dizziness or lost consciousness within the past 12 months; (5) bone, joint, or soft tissue problem that could be made worse by becoming more physically active; (6) physical activity need to be medically supervised. People with diseases (1) and (2) were excluded for the safety of the electrical stimulation test. To exclude the influence of participants’ other diseases on the results, we added the exclusion criteria (3), (4), (5), and (6). This project was approved by the local Institutional Review Board (H20076). All participants signed the Informed Consent Form before testing. Monofilaments tested plantar pressure sensitivity on both feet [20]. Balance was tested in a standing position. The H-reflex of lateral gastrocnemius (LG) [21] was evoked in the standing position.


**Plantar pressure sensitivity test**


Both feet’ plantar pressure sensitivity was assessed with 5.07-gauge Semmes-Weinstein monofilament (North Coast Medical, Inc, Morgan Hill, CA, USA) in the supine position. Five locations for each foot: hallux, bases of first and fifth metatarsals, lateral mid-sole, and heel [22]. Test each site three times. The participants answered “yes” when they felt the pressure. The normal sensitivity of each spot was defined by two or more correct responses and got a “1” score. The abnormal sensitivity was defined by two or more incorrect responses and got a “0” score. Both plantar sensitivities were added by each site score, ranging from 0 to 10 scores.


**Postural control test**


The data were collected using the Accusway two-dimensional force plate (AMTI, Watertown, MA, USA) with 1000 Hz in the standing position. The participants looked straight, arms relaxed, hanging on the sides, and their spines in a straight position ahead with their heels 10 cm apart with 10° abducted for the 30 s [23].


**H-reflex test**


The LG was selected to test H-reflex rather than the soleus muscle because the H-reflex of LG was more reliable than the soleus in people with plantar sensitivity deficit [24]. The right side was chosen to test. We used hand-held electrodes to determine the optimal location of nerve stimulation based on the criterion that the Ia afferent nerve can be selectively stimulated at low stimulation intensity [25]. A pre-gel disposable 2 cm × 2 cm cathode replaces the hand-held electrode at selected locations in the popliteal fossa. A 5 cm × 8 cm anode was placed on the patella. The surface electromyography (EMG) electrode (Trigno Wireless EMG System; Delsys Inc., Natick, MA, USA) was attached to the belly of the LG muscle along the muscle fiber orientation. EMG in reaction to the stimulation was collected at 2000 Hz.

The H-reflex was tested when participants stood with their feet apart at shoulder width and the ankles at a neutral position. The participants’ arms were relaxed on the sides of the body, the eyes were looking straight ahead, and the spine was kept neutral. The H-reflex was evoked by a 500 μs square-pulse constant current stimulus (Digitimer model DS7A, Digitimer Ltd., Welwyn Garden City, England) [26]. Stimulation started at 5 mA and increased with a 2 mA increment, reaching a full H-reflex recruitment curve at a 10 s stimulus interval [27].


**Data process**


The two indicators based on the center of pressure data, Standard Deviation (*SD*), and Sample Entropy (SampEN), in both anterior–posterior (AP) and mediolateral (ML) directions, were calculated for each participant to assess postural control.

The *SD* was calculated using the following equation [28]:(1)$$SD=\sqrt{\frac{\sum_{i=1}^{n}{\left(\bar{x}-x_{i}\right)}^{2}}{n-1}},$$

n indicated the length of the time series; $x_{i}$ means any one of the time series; $\bar{x}$ was the mean of time series.

The *SD* of AP and ML were calculated where n = 30,000.

The *SampEN* was calculated using the following equation [29]:

(1)from a vector $X_{N}=\left\{x_{1},\;x_{2}\right.\cdots ,x_{N}\}$, two sequences of m consecutive points.
(2)$$X_{m}\left(i\right)=\left\{x_{i},\;\right.\cdots ,x_{i+m-1}\}\;and\;X_{m}\left(j\right)=\left\{x_{j},\;\right.\cdots ,x_{i+m-1}\}\;\left(i,j\;\in \left[1,N-m\right],\;i\;\neq j\right)\;$$
(2)Computed the maximum distance:
(3)$$d\left[X_{m}\left(i\right),X_{m}\left(j\right)\right]=max\left[\left|x_{i+k}\right.,\left.x_{j+k}\right|\right]$$
and then compared with tolerance r for repeated sequences counting:(4)$$d\left[X_{m}\left(i\right),X_{m}\left(j\right)\right]\leq r\left(k\in \left[0,m-1\right],r\geq 0\right)$$
where the tolerance r equals 0.1−0.2 * *SD* [29], *SD* is the standard deviation of $X_{N}$.
(3)For the sequence $X_{m}\left(i\right)$, its count is defined as $B_{i}^{m}\left(r\right)$.$B^{m}\left(r\right)$ is the average amount of $B_{i}^{m}\left(r\right)$ for $,i\;\in \left[1,N-m\right]$.$A^{m}\left(r\right)$ is the average of $B_{i}^{m+1}\left(r\right)$.(4)

$SampEN\left(m,r,N\right)=\underset{N\to \infty }{\lim}\{-\ln[\frac{A^{m}\left(r\right)}{B^{m}\left(r\right)}]\}$



$\;=-\ln\left[\frac{A^{m}\left(r\right)}{B^{m}\left(r\right)}\right]$



(5)
$$=-\ln[\frac{{\left(N-m-1\right)}^{-1}\sum_{i=1}^{N-m-1}B_{i}^{m+1}\left(r\right)}{{\left(N-m\right)}^{-1}\sum_{i=1}^{N-m}B_{i}^{m}\left(r\right)}]$$



The Sample entropy of AP and ML was calculated where *m* = 2, *r* = 0.2, and *N* = 30,000 were used.

The *H*/*M* ratio was the indicator of H-reflex.
(6)$$\frac{H}{M}\;ratio=\frac{H_{max}}{M_{max}}*100\%$$

$H_{max}$ is the maximum value of the H-wave in the recruitment curve; $M_{max}$ is the maximum value of the M-wave in the recruitment curve.


**Statistical analysis**


The Shapiro–Wilk method was used to test for normality. The one-way ANOVA was used for the normally distributed indicators. The Kruskal–Wallis (H test) was used for the non-normal distributed indicators. The Bonferroni test was used for the post hoc whenever necessary. The statistical significance threshold was set at 0.05. Small, medium and large effect sizes for the ANOVA were set at *η_p_^2^* < 0.01, 0.01 ≤ *η_p_^2^* < 0.06, and *η_p_^2^* ≥ 0.06 [30]. Cohen’s d was defined by 0.20 ≤ d < 0.50, 0.50 ≤ d < 0. 80, and d ≥ 0.80 as small, medium, and large effect sizes for the post hoc comparisons [31]. Adj. R^2^ was used to estimate the strength of the correlation coefficient. The very weak, weak, moderate, strong, and very strong correlations were 0.00 ≤ R^2^ < 0.20, 0.20 ≤ R^2^ < 0.40, 0.40 ≤ R^2^ < 0.60, 0.60 ≤ R^2^ < 0.80, R^2^ ≥ 0.80 [32].

## 3. Results

Twenty-seven participants were divided into the Control group (10 scores), the Less affected group (6–9 scores), and MA (0–5 scores) based on the plantar pressure sensitivity test.

The different body masses were observed among the three groups (F_2,26_ = 4.783, *p* = 0.018, partial *η*^2^ = 0.285). People in the MA group were heavier than people in the Control group (significant difference in body mass, *p* = 0.005, d = 1.347). The heights of people in the three groups were also different (F_2,26_ = 4.180, *p* = 0.028, partial *η*^2^ = 0.258). The MA participants were taller than the Control participants (significant height difference, *p* = 0.009, d = 1.382). However, no BMI difference was detected. See Table 1 for more details.

The results of the normality tests were as follows. The standard deviation in AP and ML directions, the sample entropy AP and the standing H/M ratio conformed to the normal distribution (*p* > 0.05). The sample entropy ML directions did not conform to the normal distribution in the control group (*p* = 0.018). See Table 2 for more details.

Significant SampEN_AP_ in the three groups was detected (F_2,26_ = 5.232, *p* = 0.013, partial *η*^2^ = 0.304). The results show more robust regularity in the COP of MA than that of the Control group (significant difference in SampEN_AP_, *p* = 0.011, d = 1.330). See Table 3 and Figure 1 for more details.

The standard deviation and sample entropy in AP direction were highly negatively correlated in the LA (r = −0.758 Adj. R^2^ = 0.521, *p* = 0.011) and MA groups (r = −0.741 Adj. R^2^ = 0.484, *p* = 0.022), and in ML direction in the Control (r = −0.766 Adj. R^2^ = 0.517, *p* = 0.027), LA (r = −0.894 Adj. R^2^ = 0.773, *p* < 0.001), and MA (r = −0.908 Adj. R^2^ = 0.800, *p* = 0.001). The three groups did not correlate the standard deviation and standing H/M ratio. The SampEN_ML_ of the Control group decreased with the increase in standing H/M ratio (r = −0.718, Adj. R^2^ = 0.434, *p* = 0.045). See Table 4 and Figure 2 for more details.

## 4. Discussion

We have observed no loss of plantar sensation effect on the degree of COP variability, but effects on the complexity of COP variability. Less complexity was observed in the COP of MA compared with people in Control and LA groups. The SD of COP was strongly correlated with the SampEN of COP in the AP direction in the LA and MA groups. The SD of COP was strongly related with the SampEN of COP in the ML direction among all groups. These observations support our first hypothesis. The increment of SampEN COP was related to the decrease in the standing H/M ratio in healthy people, which supports the first part of the second hypothesis.

People with severe plantar sensory deficit may have decreased standing postural control, mainly reflected in the COP complexity. Our observations were inconsistent with previous studies [13]. The results of the present study indicated that impaired plantar sensation modifies the complexity of postural swing variability, but not the degree. Hong and colleagues pointed out that the increase in postural swing amplitude was due to reduced plantar feedback. The input sources of plantar feedback information are blocked, and the CNS is not sensitive to changes in the COP position and is only correct when the COP amplitude is large. The results of this study implied that people with MA exhibited less COP complexity, that is, lower flexibility and higher stiffness for postural control [33,34] and poorer postural control [35]. The sites of plantar sensation of this study’s participants were different. There was no complete loss of plantar sensation in the LA group, and most participants had sufficient plantar sensation to maintain the structure of the COP, resulting in no difference from non-PN. This may be a reason why Hong et al. did not observe structural changes in COP. However, some participants in the MA group suffered a complete loss of plantar sensation, which affected the structure of the COP.

The degree and complexity of COP variability are strongly correlated. It is inferred that the structure of COP variability in people without PN is relatively independent in the AP and ML directions, while the people with PN are highly correlated. The property of the COP variability complexity has been validated and applied in healthy young [36,37] and older adults [34,38,39,40]. The entropy of the COP of older adults showed inconsistently in AP and ML directions and was not affected by posture in healthy young. Our study supported this notion, and the same results emerged with COP entropy in people without PN. No study has yet verified a similar pattern in people with PN. This study inferred that plantar sensation may be an important factor in maintaining the independence of COP entropy.

People with PN have adaptive changes in CNS, which changes standing posture control. It was observed in our previous study [10] that the CNS function of people with PN adapted to the loss of plantar sensation, and the H-reflex during standing between groups (elderly people with and without PN). However, H-reflex was more affected by different postures. From prone to standing, H-reflex was significantly suppressed in non-PN, but not in PN. Older adults with PN may rely more on proprioceptive sense with larger-diameter nerve fibers to maintain standing balance [41,42,43]. Nonetheless, this study goes a step further to understand that more reliance on proprioceptive sense with larger-diameter nerve fibers still cannot compensate for the imbalance caused by damage to tactile sensation with smaller-diameter nerve fibers. Moreover, the standing postural control was less modulated by the CNS in people with PN, showing that inhibition or excitation of the H-reflex did not affect the entropy of the COP, in both LA and MA groups.

In previous studies on entropy, postural control of specific population was usually studied based on age [38], physical condition [34], and falls [40]. A few scholars have studied the effect of neurological changes on posture control. The main research method was the temporary loss of plantar sensation using anesthesia injection [44] and ice compress [13]. These methods can quickly reach experimental conditions and have high controllability. They can effectively study the effect of a single factor (plantar sensation) on posture control, which has a high reference value. However, these research methods neglected that the body is a complex system, the neuromuscular system is interlinked, and changes in any one link may cause adaptive changes in other related links. Temporary plantar sensation loss is not a substitute model for people with PN suffering from impaired plantar sensation. Inspired by previous studies, the present study explored the impact of impaired plantar sensation on CNS modulation and the role of the CNS in standing posture control after long-term adaptation.

Some limitations of this study should be considered to improve future studies. A comprehensive CNS and spinal cord medical examination for potential confounding factors could strength the observation and the interpretation of the project. The complexity of COP while standing is influenced by the integration of multiple factors, including tactile sensation, vestibular sensation, proprioception, central nervous system, etc. The focus of this study was on the impact of tactile sensation and the CNS on COP complexity, so the specific mechanism underlying the altered COP complexity in people with PN is unclear. Future studies should employ a more comprehensive design to study COP complexity and explore the impact of specific factors on COP complexity.

## 5. Conclusions

(1) The complexity of COP variability in patients with severe plantar sensory impairment is changed, which will not affect the degree of COP variability; (2) The independence of the COP entropy in the AP and ML directions decreased, and the interdependence increased in people with PN; (3) Although the CNS of people with PN has a greater contribution to standing balance, its modulation of standing postural control is decreased.

## Figures and Tables

**Figure 1 entropy-25-00064-f001:**
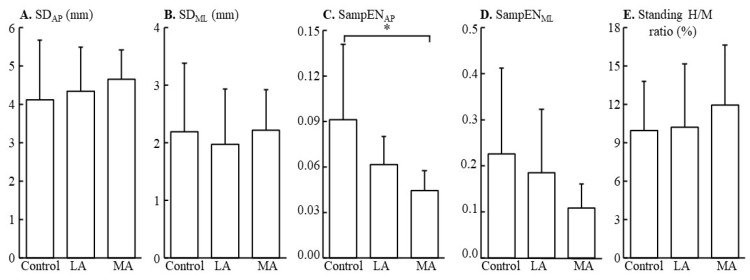
The differences in the standard deviation (**A**,**B**) in the AP and ML directions, sample entropy of AP (**C**) and ML (**D**) directions, and standing H/M ratio (**E**), among three groups (Control, LA, MA groups). * *p* < 0.05.

**Figure 2 entropy-25-00064-f002:**
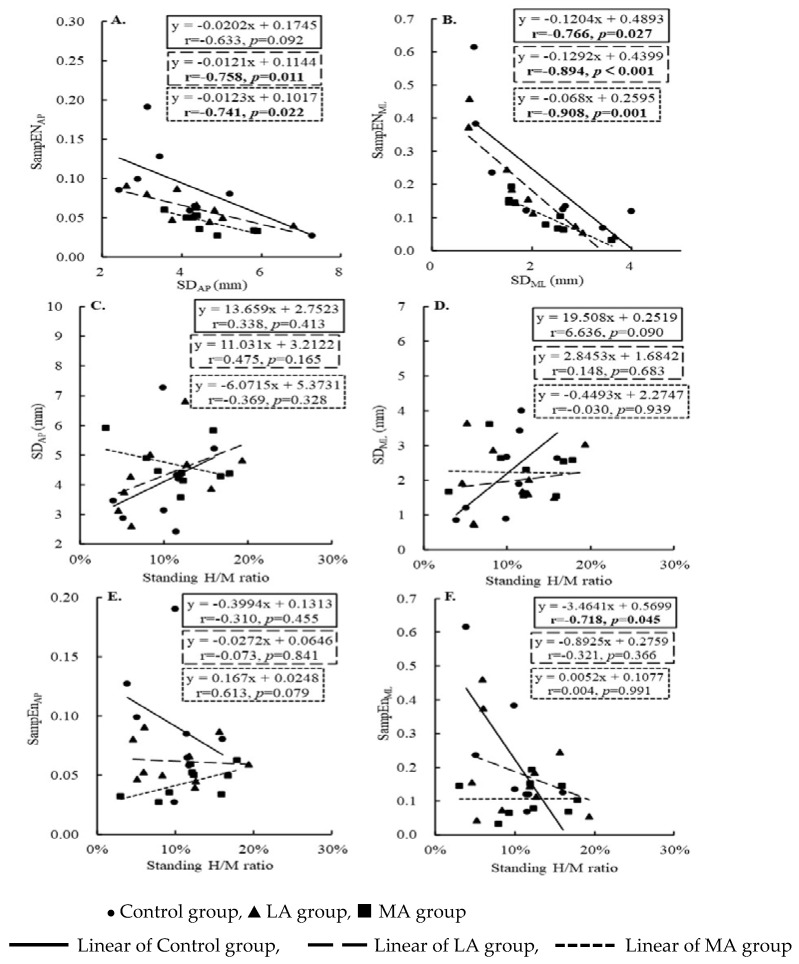
The linear relationship between the standard deviation and sample entropy (**A**,**B**), the standard deviation and H/M ratio (**C**,**D**), and the sample entropy and H/M ratio (**E**,**F**) among three groups (Control, LA, MA groups).

**Table 1 entropy-25-00064-t001:** The participants’ characteristics for the three groups.

	Control (n = 8)	Less Affected (n = 10)	More Affected (n = 9)	F_2,26_	*p*-Value
Body mass (kg)	68.1 ± 11.7	79.9 ± 12.2	91.8 ± 21.5	4.398	0.018 *
Height (cm)	162.5 ± 6.2	166.4 ± 7.7	173.1 ± 8.7	4.356	0.028 *
BMI (kg/m^2^)	25.7 ± 4.2	28.8 ± 3.9	30.4 ± 5.1	2.056	0.108

* *p* < 0.05.

**Table 2 entropy-25-00064-t002:** The normal distribution results of the Shapiro–Wilk test for three groups.

	Control		Less Affected		More Affected	
	W	*p*-Value	W	*p*-Value	W	*p*-Value
SD-AP (mm)	0.905	0.322	0.949	0.653	0.882	0.167
SD-ML (mm)	0.925	0.470	0.940	0.554	0.866	0.110
SampEN_AP_	0.933	0.545	0.904	0.239	0.914	0.343
SampEN_ML_	0.780	0.018 *	0.884	0.144	0.950	0.689
Standing H/M ratio	0.916	0.398	0.916	0.329	0.947	0.662

* *p* < 0.05 indicated that the parameter was not normally distributed.

**Table 3 entropy-25-00064-t003:** The differences in standard deviation, sample entropy in AP and ML directions, and standing H/M ratio among the three groups.

	Control	Less Affected	More Affected	*p*-Value
SD-AP (mm)	4 ± 2	4 ± 1	5 ± 1	0.648
SD- ML (mm)	2 ± 1	2 ± 1	2 ± 1	0.829
SampEN_AP_	0.09 ± 0.05	0.06 ± 0.02	0.04 ± 0.01 ^a^	0.013 *
SampEN_ML_	0.23 ± 0.19	0.18 ± 0.14	0.11 ± 0.05	0.358
Standing H/M ratio	10.0 ± 3.9%	10.2 ± 5.0%	11.9 ± 4.7%	0.619

* *p* < 0.05; ^a^: indicated have a significant difference with the Control group.

**Table 4 entropy-25-00064-t004:** The correlation between standard deviation and sample entropy, standard deviation and standing H/M ratio, sample entropy and standing H/M ratio in AP and ML directions with Control, LA, and MA groups.

	Control		LA		MA	
	r	*p*-Value	r	*p*-Value	r	*p*-Value
SD_AP_ and SampEN_AP_	−0.633	0.092	−0.758	0.011 *	−0.741	0.022 *
SD_ML_ and SampEN_ML_	−0.766	0.027 *	−0.894	<0.001 *	−0.908	0.001 *
SD_AP_ and Standing H/M ratio SD_ML_ and Standing H/M ratio SampEN_AP_ and Standing H/M ratio SampEN_ML_ and Standing H/M ratio	0.338	0.413	0.475	0.165	−0.369	0.328
	0.636	0.090	0.148	0.683	−0.030	0.939
	−0.310	0.455	−0.073	0.841	0.613	0.079
	−0.718	0.045 *	−0.321	0.366	0.004	0.991

* *p* < 0.05.

## Data Availability

Additional data are unavailable due to privacy and ethical restrictions.

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
