# Peer review of "The Impact of Hoffmann Reflex on Standing Postural Control Complexity in the Elderly with Impaired Plantar Sensation"

_entropy, 2022, doi:10.3390/e25010064_

Round 1

Reviewer 1 Report

I commend the researchers for their efforts in conducting research on interesting topics.

In the introduction

Authors should more clearly define complexity  according to their own definition, because complexity has various definitions depending on the researches. 

What is the difference between complexity of CoP variability and structure of CoP variability?

At the end of introduction, I was not able to find clear difference between hypothesis one and two. 

In materials and Methods

Exclusion criteria of subects are vague. I don't know what you mean by "(1) heart conditioning". Please provide rationale for the exclusion criteria (4), (5), and (6). 

In dicussion and conclusion

H-reflex has both components; sensory and motor which are relevant to ankle stragegy in balance control mechanism. How can you discriminate functional decrement in sensory and motor parts in H-reflex? Without CNS injury, H-reflex can be decreased or diminished by peripheral nerve injury such as S1 radiculopathy. This query makes interpretation of H-reflex for central modulation confused.

Author Response

Thank you for your very helpful comments.

Reviewer 2 Report

Please see attached

Author Response

(The authors gave the same response as above.)
